# *HUNK* Gene Alterations in Breast Cancer

**DOI:** 10.3390/biomedicines10123072

**Published:** 2022-11-29

**Authors:** Nicole Ramos-Solis, Tinslee Dilday, Alex E. Kritikos, Elizabeth S. Yeh

**Affiliations:** 1Department of Pharmacology and Toxicology, Indiana University School of Medicine, Indianapolis, IN 46202, USA; 2Melvin and Bren Simon Comprehensive Cancer Center, Indiana University School of Medicine, Indianapolis, IN 46202, USA

**Keywords:** HUNK, breast cancer, phosphorylation, metastasis

## Abstract

Hormonally upregulated neu-associated kinase (HUNK) is a serine/threonine (S/T) protein kinase related to the adenosine monophosphate-activated protein kinase (AMPK) family of kinases. HUNK was originally discovered using a screen to identify kinases expressed in the mouse mammary gland. Therefore, the majority of studies to date have been carried out in models specific to this tissue, and the kinase was named to reflect its mammary gland-specific physiology and pathology. Prior studies show a clear pathogenic role for HUNK in breast cancer. HUNK is upregulated in response to oncogenic HER2/neu and Akt, and there is strong evidence that HUNK is critical for the survival of breast cancer cells. Further evidence shows that inhibiting HUNK using a variety of breast cancer models, including those that are resistant, inhibits tumorigenesis and metastasis. However, HUNK alterations are infrequent. Here, the incidence and consequence of HUNK alterations in breast cancer is reviewed using data mined from the online database cBioPortal and considered in relation to prior research studies.

## 1. Introduction

### 1.1. Background

Hormonally upregulated neu-associated kinase (HUNK) is an approximately ~80 kDa serine/threonine (S/T) protein kinase that has been reported to be expressed in about 50% of breast carcinomas [1,2,3,4]. HUNK is part of the AMP-associated protein kinase (AMPK) family and plays a role in a variety of cellular processes, including the regulation of cell survival, proliferation, and autophagy [1,5]. While this protein was originally discovered in mouse mammary glands and is expressed in breast tissue, it has also been found in other organ systems such as the brain and central nervous system, and plays a role in diseases including colon cancer [3,4,6,7,8]. Normal mammary gland development in mice is associated with HUNK activity, and breast cancer progression and metastasis has also been attributed to HUNK [9,10,11]. Specifically in the HER2+ subtype of breast cancer, HUNK promotes HER2-targeted drug resistance due to its role in autophagy [12,13]. HUNK’s role in metastasis is limited to studies predominantly in the basal breast cancer subtype and there is a general consensus that HUNK promotes metastasis [10,11].

### 1.2. HER2-Positive Breast Cancer

HER2+ breast cancer makes up around ~15–20% of breast cancers and is associated with poor outcomes in breast cancer patients. HUNK is upregulated in HER2+ breast cancer and high *HUNK* expression is associated with poor prognosis [1,13,14]. In HER2+ breast cancer cell lines and mammary tumors, evidence suggests that HUNK regulates both autophagy and apoptosis making it a critical survival protein in HER2+ breast cancer [13]. In particular, knocking down HUNK in HER2/neu+ breast cancer models causes impaired mammary tumorigenesis, even in resistant models [13,14]. Furthermore, is also has been shown that the multi-kinase inhibitor, Staurosporine, is an inhibitor of HUNK kinase activity and synergizes with the HER2 inhibitor, lapatinib [15]. Taken together, prior studies indicate that inhibiting HUNK kinase activity shows promise in the treatment of HER2+ breast cancer, especially HER2+ breast cancers that have become refractory to HER2 inhibitors [15]. 

### 1.3. Metastasis

In 2009, Wertheim et.al was one of the first groups to describe a pro-metastatic function of HUNK in mice [10]. The study showed that *Hunk* is required for mammary tumor metastasis in MMTV-*c-myc* transgenic mice. In this study, MMTV-*c-myc;Hunk* wildtype (+/+) mice had higher levels of lung metastasis than MMTV-*c-myc;Hunk* knockout (−/−) mice. Tumor cells isolated from MMTV-*c-myc;Hunk*−/− mice that express wildtype HUNK or a kinase-deficient form of HUNK showed that HUNK kinase activity is required for tumor cell migration, invasion, and metastasis. Contrary to the previous study, a second study published in 2009 by Quintela-Fandinoa et.al showed that HUNK suppresses metastasis in basal breast cancer [16]. This second study showed in MDA-MB-468 and MDA-MB-231 cells, that HUNK over-expression impaired cell migration, invasion, and metastasis [16]. More recent studies by Williams et al. in 2020, showed that HUNK promotes breast cancer metastasis by phosphorylating the epidermal growth factor receptor (EGFR) at threonine (T) 654, enhancing downstream signaling from the receptor [11]. Using BT20, MDA-MB-468, and 4T1 tumor cell lines, this study showed that depleting HUNK by shRNA knockdown impaired migration, invasion, and metastasis similar to the study by Wertheim et al. [10,11].

Overall, very few studies have been published addressing HUNK’s signaling activities and downstream substrates. Few direct substrates for HUNK have been discovered but of the handful that are described, a role for HUNK in autophagy and regulation of metastatic signaling is clear [11,15,17]. This review article will provide an analysis of what is currently known about HUNK substrates and signaling, and will address new findings about *HUNK* alterations in cancer, with focus on breast cancer. 

## 2. *HUNK* Alterations in Breast Cancer

### 2.1. Incidence and Mutations

Since the protein was discovered, there has been mounting evidence of HUNK’s role as a pro-tumorigenic protein, particularly in breast cancer [1,2,9,10,11,12,13,14,16]. However, HUNK as a single aberrantly active protein is not pathogenic unless a second pro-tumorigenic factor is also present in an oncogenic form (i.e., HER2, AKT). Long-term HUNK overexpression in the breast does not lead to spontaneous tumor formation [4]. On the contrary, *Hunk* genetic deletion in mice or HUNK targeting (i.e., knockdown or kinase targeting) impairs tumor growth and incidence in response to oncogene activation [9,12,13,14]. Due to the historically important recognition of oncogenes and tumor suppressors, it is a popular, but unfortunately false notion that all important pro-cancer proteins are genetically altered at high frequency. The infamous mammalian Target of Rapamycin (mTOR) protein kinase comes to mind. mTOR is pro-tumorigenic but not frequently mutated or altered (i.e., deleted, amplified, gene-fusion) [18]. Yet, a significant amount of energy has been expended on successfully “drugging” this kinase for cancer treatment and mTOR’s relevance in cancer pathogenicity is rarely in question. 

HUNK, on the other hand, is not well known and has not been studied extensively in all tissue types. Very few substrates and functions are identified for HUNK but to date the majority of downstream signaling consequences clearly indicate a pro-tumor, cancer supportive function for this kinase [11,15,17]. Reports of *HUNK* gene alterations and frequency of alteration are absent from the literature. Therefore, we undertook this analysis using the popular online data mining tool, cBioPortal [19,20]. First, a pan-cancer search of alteration frequency focused on two major cancer kinases, mTOR and AMPK, was conducted and compared to HUNK. This search included all datasets available in cBioPortal to provide an overarching view of alteration frequency in multiple human cancers. The results from this search indicated that there is a frequency of mutation less than 3% for each of these kinases across all types of human cancers. 

We then looked at the breast cancer datasets in isolation from the other cancer types. These datasets contain a combined 11253 samples from 24 studies. In breast cancer, there is less than a 2% mutation frequency for mTOR (1.9%), AMPK (1.8%), and HUNK (1.3%). The analysis presented in Figure 1 focuses on HUNK. Figure 1A shows the alteration types and frequencies found in HUNK for each of the breast cancer datasets analyzed that included alteration information. Figure 1B illustrates the specific alterations identified (see also Appendix A). Among these alterations, four copy number gains, one shallow deletion, one nonsense mutation, and a number of missense mutations were observed. In addition, there were two gene fusions identified: *HUNK-MRAP* and *EVA1C-HUNK.* Perhaps unsurprisingly, *MRAP* (Melanocortin 2 Receptor Accessory Protein) and *EVA1C* (Eva-1 Homolog C) are located in close proximity to *HUNK* on chromosome 21. Among the missense mutations, several were identified in the kinase domain of HUNK. The remainder were in the C-terminal portion of the protein, which is less well characterized but presumed to be involved in protein–protein interaction [1]. 

### 2.2. Outcomes

Experimental evidence to date mainly indicates that HUNK over-activity or over-expression is associated with poorer outcomes for breast cancer [1,2,9,10,11,12,13,14,16]. Prior studies show that high *HUNK* expressions correlates with *HER2* amplification in breast cancer and poor prognosis [1]. Furthermore, gene expression analysis shows that a HUNK signature is associated with more aggressive breast cancer subtypes [10]. Experimental studies showing that HUNK promotes tumor progression of HER2/neu+ breast cancer and metastasis of basal-type breast cancer confirm the gene expression analyses [10,11,13,14]. 

Consistent with prior published findings, breast cancer samples from patients with alterations in *HUNK* were associated with poorer outcomes. Shown here, median time of overall survival (Figure 1C) and relapse free survival (Figure 1D) are significantly reduced in *HUNK* altered groups.

## 3. Pathway Analysis

More recently, HUNK’s intracellular functions have begun to be uncovered, however, the protein is still relatively under-characterized. Similar to the false assumption that pro-tumor proteins typically have high gene alteration levels in cancer, there is the artificial consideration that kinases have only one important target substrate that carries out their pro-cancer functions. Phospho-proteome analysis of mTOR and AMPK reveal numerous substrates, many of which are associated with potential pathogenic functions of these kinases [21,22]. 

To date there are only a handful of substrates and a few intracellular functions that are described for HUNK [11,15,17]. Of note, HUNK substrates are predominantly involved in cancer promoting functions and therefore, targeting HUNK is expected have an overall anti-cancer effect when considered in the context of the downstream consequences of modifying HUNK substrates. Prior studies show that HUNK has a pro-autophagy function and inhibiting this function is anti-tumorigenic specifically in the context of HER2/neu+ breast cancer [12,13,15,23]. Although autophagy has both pro- and anti-tumorigenic functions, in the context of breast cancer, autophagy is typically cyto-protective [24]. Whether HUNK’s autophagy function plays a role in other breast cancer subtypes remains to be determined. Of the handful of HUNK substrates that have been identified, the majority are implicated in regulation of autophagy [15,17]. HUNK has also been shown to phosphorylate EGFR and inhibiting HUNK results in a loss of EGFR phosphorylation that is associated with an anti-metastatic effect [11]. Beyond these few studies, very little is known about HUNK intracellular signaling. 

Extensive analysis of HUNK signaling using “omic” strategies has not been undertaken. Recently, researchers have integrated the data from the Clinical Proteomic Tumor Analysis Consortium (CPTAC) into cBioPortal [25]. This undertaking facilitates comparing mRNA, proteomic, and phospho-proteomic data for a subset of breast tumors (122 total in this analysis) from The Cancer Genome Atlas (TCGA). To explore the impact of *HUNK* alterations on cancer signaling, we took advantage of this cBioPortal feature. Figure 2 shows volcano plots exemplifying the significant changes in mRNA (Figure 2A), protein (Figure 2B), and phosphorylated proteins (Figure 2C) from *HUNK* altered and unaltered samples. A complete list of factors is reported in Appendix A. The following sections discuss in further detail the pathway related predictions associated with these differences. 

### 3.1. mRNA

The genomic data from CPTAC were analyzed for pathway enrichments in either HUNK altered or unaltered samples. There were 249 gene transcripts that were higher in the HUNK altered group and 118 genes that were lower (Appendix A). mRNAs identified in this dataset were sorted by false discovery rate (FDR) using the most significant *p*-value as the primary indicator. The STRING consortium resource was used to examine signaling pathways based on gene representation that were either over-represented or under-represented in the *HUNK* altered group. Data files for the STRING analysis are provided as Appendix A. 

Factors that were over-represented in the *HUNK* altered group, were assessed using Gene Ontology (GO) Biological Processes, Kyoto Encyclopedia of Genes and Genomes (KEGG), and Reactome databases that were consolidated by the STRING analysis. The prediction by each of these databases implies that the over-represented mRNAs in the *HUNK* altered samples are indicative of changes in immune system regulation (Table 1). Of note, the KEGG and Reactome analyses indicate that T-cell, cytokine, and chemokine signaling among other biological processes are predicted to be significantly affected by *HUNK* alteration (Table 1 and Appendix A). This observation was confirmed by GO molecular function analysis, which identified four networks that contained mRNAs that were over-represented in the *HUNK* altered list; Cytokine receptor activity, T-cell receptor binding, Chemokine receptor binding, and CCR chemokine receptor binding (see Appendix A). 

In confirmation of prior work in breast cancer, an analysis of reference publications within PubMed containing co-occurring mention of the group of mRNAs higher in *HUNK* altered samples showed a significant association with breast cancer. The PubMed reference publication analysis also showed publications that reinforce the new finding reported here that pathways associated with immune cell signaling are potentially affected by *HUNK* alterations (see Appendix A).

Interestingly, when the analysis was performed on mRNAs that were found in the lower in *HUNK* altered group, there was a very minimal number of pathway interactions identified. GO biological processes analysis only yielded one association: homophilic cell adhesion via plasma membrane adhesion molecules (Table 1). GO molecular function analysis suggested an association of genes with calcium ion binding (Appendix A). The under-representation of the biological process of cell-to-cell adhesion in *HUNK* altered samples fits with prior reports that indicate HUNK has a pro-metastatic function (Appendix A) [10,11].

### 3.2. Proteome

The total proteome data from CPTAC showed that among the >9000 proteins identified, 287 proteins were significantly higher and 583 proteins were significantly lower in the *HUNK* altered group compared to the unaltered group (Figure 2B). As with the mRNA analysis, proteins were sorted by false discovery rate (FDR) using the most significant *p*-value as the primary indicator. The STRING consortium resource was then used to examine signaling pathways with proteins that were either significantly over-represented or under-represented in the *HUNK* altered group [26]. Data files are provided as Appendix A. 

The prediction by each of the databases searched including GO biological processes, KEGG, and Reactome, again implies that the over-represented proteins in the *HUNK* altered samples are indicative of changes in immune system regulation (Table 2 and Appendix A). This finding in both the mRNA and the protein dataset was somewhat surprising as HUNK has not been extensively described as an immune regulatory kinase. A PubMed search yielded only one publication implicating HUNK in a cancer immune function. This study suggests that *HUNK* is one of eight genes that can be used as a prognostic signature for diffuse large B-cell lymphoma [25]. STRING local network cluster analysis also showed a strong correlation with ribosome biogenesis in the *HUNK* altered group (Appendix A). This observation was not seen in the mRNA analysis and was also somewhat surprising as ribosome biogenesis has not been described as a HUNK intracellular function.

The proteins identified that were lower in *HUNK* altered samples were also evaluated. GO, KEGG, and Reactome pathway analysis indicated significant changes in proteins involved in extracellular matrix (ECM) organization as well as cell-to-cell contact related processes (Table 2 and Appendix A). Again, these observations are consistent with the two prior reports that indicate a role for HUNK in promoting breast cancer metastasis [10,11]. 

### 3.3. Phospho-Proteome

There are few HUNK substrates that have been identified and HUNK-related phospho-proteomic studies are very limited. Presently, the phospho-proteomic data from CPTAC were examined. Of the subset that represented statistically significant events in the more than 30,000 changes that were detected, there were 1219 events detected that had higher levels of phosphorylation in the *HUNK* altered group and 1164 that had lower levels of phosphorylation (Appendix A). Of note, UVRAG and Rubicon, two HUNK interacting proteins appeared in the events listed in the over-represented in *HUNK* altered group. Rubicon is a HUNK substrate but interestingly, neither of the Rubicon phosphorylation sites identified in the CPTAC data are previously identified HUNK phosphorylation sites [15]. Overall, there is a significant void of information examining Rubicon phosphorylation.

UVRAG has not been reported to be phosphorylated by HUNK but resides in complex with HUNK and Rubicon [15]. One of the UVRAG phosphorylation events identified in the higher in *HUNK* altered group is at serine (S) 498 (Appendix A). Prior studies showed that this site is phosphorylated by mTOR and is required for the Rubicon-mediated inhibition of autophagy through Vps34 [27]. This observation is contradictory to prior findings that HUNK has a pro-autophagy function but has potential to indicate that mTOR function works in opposition to HUNK at the Beclin-1/Vps34 autophagy regulatory complex. UVRAG also appeared in the list of phosphorylated proteins that were under-represented in the *HUNK* altered group. In this case, UVRAG S509 phosphorylation is under-represented in the *HUNK* altered group (Appendix A). This site is reported to be phosphorylated by AKT and leads to dissociation of the UVRAG associated Beclin-1/Vps34 complex, which inhibits autophagy [28]. This relationship affirms prior reports that indicate HUNK has a pro-autophagy function. 

Another protein of interest that was identified as having a significant phosphorylation event in the phospho-proteomic data is HER2 (aka ERBB2). Phosphorylation of HER2 at two sites, T1132 and S1151, was identified in the list that was under-represented in the HUNK altered group. While these residues reside in the C-terminal intracellular domain of HER2 and would suggest increased intracellular signaling, reports of phosphorylation events at these sites are lacking. This observation is potentially interesting as HUNK was reported to be upregulated downstream of HER2, which suggests that HER2 should be more highly active when HUNK is more highly active [14]. However, there is a possibility that subsequent downregulation of HER2 activity occurs in response or as a consequence of HUNK over-activation given that HUNK promotes resistance in HER2+ breast cancer [13]. 

Pathway analysis of the factors with significant changes in phosphorylation status from the phospho-proteome data generally affirmed prior observations with HUNK. GO, KEGG, and Reactome analysis indicated significant association with organelle organization, endocytosis, and cell cycle, respectively (Table 3 and Appendix A). In the set of proteins significantly altered by phosphorylation that were under-represented in the *HUNK* altered group there was a significant decrease observed in cellular component organization, tight junction, and Rho GT29ase cycle observed in the GO, KEGG, and Reactome analyses (Table 3 and Appendix A). It is also worth mentioning that Reactome analysis showed changes in death receptor signaling and apoptosis, two known functions of HUNK, suggesting that reduced phosphorylation of proteins in *HUNK* altered samples results in lower signaling through these pathways (Appendix A). Other interesting observations include an association with EGF/EGFR signaling pathway identified by the tool WikiPathways and an association with mental health disorders in the disease-gene association analysis (Appendix A).

## 4. Conclusions and Future Perspectives

The present analysis of CPTAC data supports prior studies and adds new insight into HUNK’s pro-cancer activities. However, studies using experimental models are needed to determine whether new insights are founded. Although only a small number of substrates have been identified for HUNK, these substrates predominantly have a pro-cancer effect, supporting a pro-cancer role for HUNK. The idea that HUNK has more than one substrate is not out of the norm, since having multiple substrates has not prevented other pro-tumor kinases from being pursued as anti-cancer targets. Future studies should address whether the *HUNK* mutations identified in this study change HUNK enzymatic activity and interactome. Further studies in other tissue types are also needed to determine if HUNK function is universal in each tissue. 

## Figures and Tables

**Figure 1 biomedicines-10-03072-f001:**
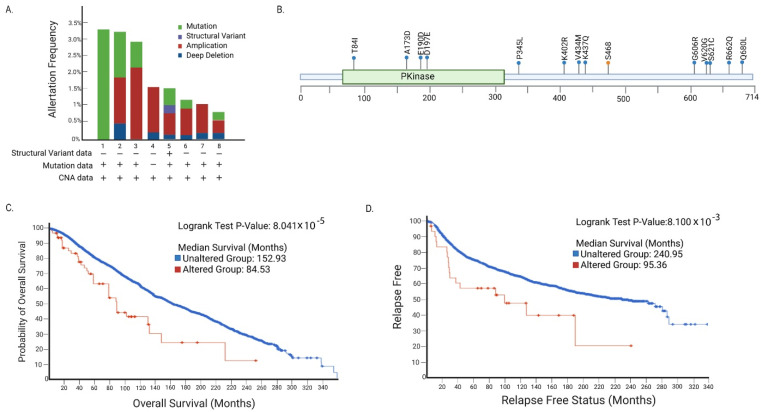
**HUNK alteration in cBioPortal breast cancer datasets.** (**A**) *HUNK* appeared in 8 of 24 studies and has an alteration frequency of up to 3% (1: Proteogenomic landscape of breast cancer (CPTAC), 2: Metastatic Breast Cancer (INSERM), 3: The Metastatic Breast Cancer Project, 4: Breast Cancer (METABRIC), 5: Breast Invasive Carcinoma (TCGA, PanCancer Atlas), 6: Breast Invasive Carcinoma (TCGA, Firehose Legacy), 7: Breast Invasive Carcinoma (TCGA), 8: Breast Invasive Carcinoma (TCGA). (**B**) Specific alterations that are identified in HUNK (each dot represents a single mutation, blue dot = missense mutation, orange dot = nonsense mutation). (**C**) Kaplan–Meier analysis showing patients with alterations in *HUNK* have poorer outcomes indicated by median time of overall survival, and (**D**) relapse free survival shown.

**Figure 2 biomedicines-10-03072-f002:**
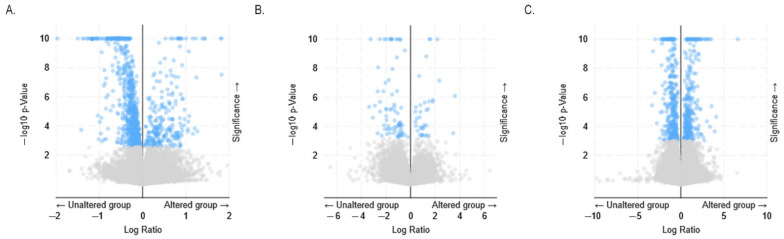
**HUNK alterations on cancer signaling.** (**A**) Using the data from CPTAC (Clinical Proteomic Tumor Analysis Consortium) in cBioPortal, 122 breast tumors from TCGA (The Cancer Genome Atlas) were examined for significant changes of mRNA, shown as a volcano plot. (**B**) Volcano plot showing significant changes in protein. (**C**) Volcano plot showing the significant changes in phosphorylated proteins.

**Table 1 biomedicines-10-03072-t001:** mRNA analysis: Factors that were over-represented in the HUNK altered group assessed using GO Biological Processes, KEGG, and Reactome databases that were consolidated by the STRING analysis.

Pathway (Higher in HUNK Altered)	Observed Gene Count	Background Gene Count	False Discovery Rate
**GO Biological Processes**			
Immune system process	90	2481	5.53 × 10^−20^
Immune response	63	1588	1.98 × 10^−14^
**KEGG**			
T cell receptor signaling pathway	12	101	1.16 × 10^−6^
Chemokine signaling pathway	15	186	1.32 × 10^−6^
Cytokine–cytokine receptor interaction	16	282	2.50 × 10^−5^
**Reactome**			
TCR signaling	12	112	7.21 × 10^−6^
Chemokine receptors bind chemokines	8	68	0.00021
Cytokine Signaling in Immune system	23	681	0.0012
**Pathway (lower in HUNK altered)**			
**GO Biological Processes**			
Homophilic cell adhesion via plasma membrane adhesion molecules	15	164	5.44 × 10^−10^

**Table 2 biomedicines-10-03072-t002:** Proteome analysis: Factors that were over-represented in the HUNK altered group assessed using GO Biological Processes, KEGG, and Reactome databases that were consolidated by the STRING analysis.

Pathway (Higher in HUNK Altered)	Observed Gene Count	Background Gene Count	False Discovery Rate
**GO Biological Processes**			
Immune system process	98	2481	2.10 × 10^−16^
Immune response	68	1588	4.20 × 10^−12^
**KEGG**			
T cell receptor signaling pathway	11	101	2.88 × 10^−5^
Chemokine signaling pathway	9	186	0.0229
**Reactome**			
TCR signaling	17	112	3.93 × 10^−9^
Cytokine Signaling in Immune system	33	681	1.82 × 10^−6^
**Pathway (lower in HUNK altered)**	Observed gene count	Background gene count	False discovery rate
**GO Biological Processes**			
Extracellular matrix organization	47	338	7.73 × 10^−13^
Cell adhesion	75	925	9.36 × 10^−11^
**KEGG**			
ECM–receptor interaction	16	88	8.13 × 10^−6^
**Reactome**			
Extracellular matrix organization	48	301	4.29 × 10^−16^

**Table 3 biomedicines-10-03072-t003:** Phospho-proteome analysis: Factors that were over-represented in the HUNK altered group assessed using GO Biological Processes, KEGG, and Reactome databases that were consolidated by the STRING analysis.

Pathway (Higher in HUNK Altered)	Observed Gene Count	Background Gene Count	False Discovery Rate
**GO Biological Processes**			
Organelle oragnization	285	3450	1.28 × 10^−19^
**KEGG**			
Endocytosis	26	241	0.0419
**Reactome**			
Cell cycle	62	647	0.00023
**Pathway (lower in HUNK altered)**	Observed gene count	Background gene count	False discovery rate
**GO Biological Processes**			
Cellular component organization	403	5447	1.41 × 10^−32^
**KEGG**			
Tight junction	25	156	2.41 × 10^−5^
**Reactome**			
Rho GTPase cycle	25	152	1.70 × 10^−6^

## Data Availability

Data mined from cBioPortal (https://www.cbioportal.org/) was accessed on 6 September 2022 and presented in Appendix A.

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
