# Peer review of "HUNK Gene Alterations in Breast Cancer"

_biomedicines, 2022, doi:10.3390/biomedicines10123072_

Round 1

Reviewer 1 Report

The manuscript reviews the biological effects and clinical outcomes of the HUNK gene (Hormonally Upregulated Neu-associated Kinase) in breast cancer published in articles which used microarray analyses to evaluate the effects of alterations on HER-2 positive breast cancer, metastasis, mutations, overexpression associated with poorer outcomes for breast cancer, pathway analysis, transcriptome, proteome, phospho-proteome. Authors stress also that alterations in HUNK gene are not very frequent. They show that the comparison between the group of breast cancer patients with alteration in HUNK gene versus control group influences expression of the other genes on the level of transcription, proteome, phopho-proteome.

The manuscript is interesting and it gives the reader information that the mutation in one gene can modify the activity of many others.

Author Response

We thank the reviewer for their supportive critique of the manuscript.

Reviewer 2 Report

In their manuscript, the authors reviewed HUNK alterations in breast cancer. The authors confirmed previous reported results and found HUNK alterations to be associated with poorer outcomes. In addition, the authors identified new pathways and new targets affected (higher and lower expression) by HUNK alteration.

The manuscript is very well written and the results are clearly presented. However, the authors should consider transforming this manuscript into a research article instead of a review and they should include a method section.

Minor comment: Fig. 1B: the colors of the dots should be brighter. Grey and green might appear similar to color blind readers.

Author Response

We thank the reviewer for their supportive critique of the manuscript. Given the limited scope of the document, we did not consider it for a research article format but appreciate the suggestion.

As suggested, we have updated Figure 1B to represent a more inclusive color schematic. We kindly thank the reviewer for providing this input.

Round 2

Reviewer 2 Report

In their manuscript, the authors reviewed HUNK alterations in breast cancer. The authors confirmed previous reported results and found HUNK alterations to be associated with poorer outcomes. In addition, the authors identified new pathways and new targets affected (higher and lower expression) by HUNK alteration.

The manuscript is very well written and the overall presentation of the initial manuscript has been improved. Previous comments have been addressed. No more comment needs to be addressed.